# Impairment in the Intestinal Morphology and in the Immunopositivity of Toll-like Receptor-4 and Other Proteins in an Autistic Mouse Model

**DOI:** 10.3390/ijms23158731

**Published:** 2022-08-05

**Authors:** Caterina Franco, Marzia Gianò, Gaia Favero, Rita Rezzani

**Affiliations:** 1Anatomy and Physiopathology Division, Department of Clinical and Experimental Sciences, University of Brescia, 25123 Brescia, Italy; 2Interdipartimental University Center of Research “Adaption and Regeneration of Tissues and Organs-(ARTO)”, University of Brescia, 25123 Brescia, Italy; 3Italian Society of Orofacial Pain (SISDO), 25123 Brescia, Italy

**Keywords:** autism spectrum disorder, BTBR mice, gastrointestinal diseases, microbiota, Toll-like receptor 4, pro-inflammatory and apoptotic proteins, oxidative stress

## Abstract

Autism spectrum disorder (ASD) identifies a neurodevelopmental disease defined by social impairments and repetitive or stereotyped behaviors. The etiology of ASD remains unclear; it primarily affects the brain, but a link between gastrointestinal (GI) diseases, inflammatory mucosal pathology and this disorder has been suggested. In particular, a central role seems to be played by an imbalance in pro-and anti-inflammatory cytokines, oxidative stress, and apoptosis. Toll-like receptor 4 (TLR4) is a protein of innate immunity responsible for the regulation and maintenance of intestinal homeostasis. Through histochemical and immunohistochemical evaluations we analyzed the intestinal morphology and the immunopositivity of TLR4 and of other pro-inflammatory and apoptotic proteins in BTBR T+Itpr3tf/J mice. Morphological data showed that the mucosal tunica presented longer intestinal villi. The length of the villi and the epithelial surface determine the exchanges of the intestinal mucosa with luminal contents, modifying the microbiota composition. The biochemical and immunohistochemical results indicated a close relationship among the increase of TLR4 and the activation of NF-kB subunits (p65 and p50) and pro-inflammatory and apoptotic proteins, such as cyclooxygenase-2, interleukin-1β, inducible nitric oxide synthase, tumor nuclear factor—alpha, caspase-3, caspase-8. These preliminary results require more in-depth study but they suggest the TLR4 signaling pathway as a possible target for therapeutic approaches to reduce GI disorders in ASD.

## 1. Introduction

Autism spectrum disorder (ASD) is a multifaceted neurodevelopmental disease defined by social impairments and the presence of repetitive or stereotyped behaviors [1,2,3,4]. Although ASD is primarily considered a condition that affects the brain, many diseases in other organs have been observed [5,6,7,8,9].

The etiology of ASD remains unclear, although there are suggestions of an association between ASD and inflammatory mucosal pathology, and, recently, a link also been suggested between gastrointestinal (GI) problems and symptoms in autistic patients [10]. Moreover, many studies have also shown that most autistic children have GI disorders such as chronic diarrhea, abdominal pain, vomiting, gastroesophageal reflux and altered intestinal integrity [11,12,13,14,15]. Last but not least, many researchers recently have focused their attention on the role that the enteric nervous system (ENS) could have in determining GI symptoms [16].

Despite their prevalence, GI disorders are often overlooked [17,18].

Several lines of evidence point to ongoing inflammation as responsible in the pathogenesis/progression of ASD [4,19,20]. The authors have demonstrated that autistic children show an imbalance in pro-inflammatory and anti-inflammatory cytokines both in the brain, in the periphery and in lymphatic cells themselves [21,22]. Other studies have suggested that oxidative stress and apoptosis also play a central role in this disorder, showing increased levels of cellular stress and lower antioxidant capacity [23]. In detail, oxidative stress has been studied at the membrane level and by measuring products of lipid peroxidation, and antioxidants involved in the defense system against reactive oxygen species (ROS) [24]. In addition, caspases are related to several proinflammatory processes [25] and play a vital role in the induction and amplification of intracellular apoptotic signals [26].

Toll-like receptor 4 (TLR4) is a highly conserved protein of innate immunity that is responsible for the regulation and the maintenance of homeostasis [27]. It is widely expressed on the surface of many cells throughout the human body, including immune cells, neurons and epithelial cells along the GI tract [28,29]. Although TLR4 immunopositivity and signaling are vital in healthy states contributing to the maintenance of the intestinal barrier, the recognition and response to invading pathogens and gastric mobility [30,31] mean this receptor is also involved in the development and progression of many inflammatory pathologies, particularly those of the GI tract [32,33].

Moreover, the increase of TLR4 levels is also important for the pathogenesis of ASD as demonstrated by Ahmad et al. (2018) in an autistic animal model (BTBR T+Itpr3tf/J or BTBR mice) [34]. TLR4 acts through the production of inflammatory cytokines/chemokines and ROS [35,36,37], playing an important role in the progression of neuroinflammation observed in ASD [37,38,39]. Burgueno et al. (2021) showed a relationship between up-regulated epithelial TLR4 signaling, the microbiota and the development of tumorigenesis. The microbiota activates TLR4, which in turn stimulates epithelia ROS, inducing a *closed network* for tumor progression [40].

It is known that TRL4 activates nuclear factor-kB (NF-kB) signaling pathways linked to the synthesis of many inflammatory proteins such as inducible nitric oxide synthase (iNOS), cyclooxygenase-2 (COX-2), tumor necrosis factor-alpha (TNF-α) and interleukins, such as interleukin-1β (IL-1β), in many human diseases, including cardiovascular diseases, diabetes, bipolar disorder and chronic fatigue syndrome [21,35,41].

The association between immune alteration and GI disorders in ASD is not well understood; however, BTBR mice provide a well-established model to investigate these problems. These mice show low reciprocal social interactions, repetitive behaviors such as cognitive inflexibility, increased grooming, impaired juvenile play and poor social approach [20,21,34]. Previously, our team [15] and other groups [42] have demonstrated that these mice have alterations in intestinal mucosal cells and a different presence of the sodium/glucose transporters. These results suggest a probability of becoming overweight or even obese, conditions that are comorbidities in autistic patients.

The aim of this study was to test by morphological and biological studies in BTBR mice the role of TLR4 and some proinflammatory proteins such as NF-kB (and its subunits: p50 and p65), iNOS, COX-2, TNF-α and IL-1β in GI disorders, which are symptomatic, as reported above, in autistic patients. Moreover, we also studied the apoptotic mechanisms by caspases enzymes.

This research could be a starting point for evaluating the morphology of the small intestine in BTBR mice and assessing the role of TLR4 and other related proteins in a new therapeutic approach for treating the comorbidities associated with ASD.

## 2. Material and Methods

### 2.1. Experimental Groups

Twenty BTBR T+Itpr3tf/J (JAX™ Mice Strain) mice—as a transgenic animal model of ASD—and twenty C57BL6/J (JAX™ Mice Strain) mice—as healthy control (CTR) mice—starting from an age of 3 weeks were housed in cages (two or three animals/cage) with food and water ad libitum and kept in an animal house at a constant temperature of 20 °C with a 12 h alternating light–dark cycle, to minimize the circadian variations.

Before the beginning of the experiment, mice were left housed in the animal facility for 1 week. All efforts were made to minimize animal suffering and the number of animals used. All of the experimental procedures were approved by the Italian Ministry of Health (n° 446/2018-PR) and followed the National Institutes of Health guide for the care and use of laboratory animals (NIH Publications No. 8023, revised 1978). Each mouse at the age of 13 weeks was deeply anesthetized (isoflurane 5%) and transcardically perfused with saline followed by 1 L of 4% paraformaldehyde in phosphate buffer saline (0.1 M, pH 7.4). For morphological, immunohistochemical and ultrastructural evaluation, the gut was carefully removed from each mouse [3,15].

### 2.2. Sample Processing

The small intestine (i.e., jejunum) samples were rinsed in physiological solution and dehydrated in graded ethanol, and then embedded in paraffin wax using an automatic processor Donatello series 2 (Diapath S.p.A., Bergamo, Italy).

### 2.3. Morpho-Histological Assessment

Serial paraffin sections (5 μm thick) of each sample were cut with a semiautomatic microtome Galileo semi-series 2 (Diapath S.p.A., Bergamo, Italy). Alternate sections were deparaffinized, rehydrated, and stained with Hematoxylin-Eosin (Bio Optica, Milan, Italy) according to the standard procedure and then were observed with an optical light microscope (Olympus BX50 microscope, Hamburg, Germany, RRID:SCR_018838) at a final magnification of 100×. Digitally fixed images of mucosal tunica were analyzed using an image analyzer (Image Pro-Plus, Milan, Italy, RRID:SCR_016879).

### 2.4. Immunohistochemical Evaluation

Sections from the gut of BTBR and C57BL/6 mice were cut (5 μm thick) and studied using immunohistochemical methods to evaluate TLR4, NF-kB p65, NF-kB p105/p50, COX-2, iNOS, TNF-α, IL-1β, Caspase 3 (Cas-3) and Caspase 8 (Cas-8) immunopositivity. Serial sections were deparaffinized in xylene using serial alcohol solutions and rinsed with distilled water. Sections were subjected to antigen retrieval in 0.01 M sodium citrate buffer, pH 6.0, in a microwave oven: two cycles of 3 min at 600 Watts [43]. They were later washed in Tris-buffered saline (TBS) for 5 min and incubated in 3% hydrogen peroxide for 10 min at room temperature. For showing the specificity of antibodies we used a preadbsorption test (blocking agent): 1% bovine serum albumin in 0.05% Tween 20 for 1 h at room temperature [44] and finally we incubated the sections for 40 min at 37 °C and for 1 h at room temperature with the following primary antibodies: mouse monoclonal anti-TLR4 (diluted 1:150; sc-293072; Santa Cruz Biotechnology Inc., Dallas, TX, USA), rabbit polyclonal anti-NF-kB p65 (diluted 1:100; ab16502; Abcam, Cambridge, UK), rabbit polyclonal anti-NF-kB p105/p50 (diluted 1:50; ab7971; Abcam, Cambridge, UK), rabbit polyclonal anti-COX2 (diluted 1:50; 160106; Cayman Chemical, Ann Arbor Michigan, MI, USA), rabbit polyclonal anti-iNOS (diluted 1:100; 610332; BD Biosciences, Becton, Dickinson, UK), mouse monoclonal anti-TNF-α (diluted 1:50, sc-52746; Santa Cruz Biotechnology Inc., Dallas, TX, USA), mouse monoclonal anti-IL-1β (diluted 1:50, Santa Cruz Biotechnology Inc., Dallas, TX, USA), mouse polyclonal anti-Cas-3 (diluted 1:50, sc-70497; Santa Cruz Biotechnology Inc., Dallas, TX, USA) and rabbit polyclonal anti-Cas-8 p18 (diluted 1:50, sc-7890; Santa Cruz Biotechnology Inc., Dallas, TX, USA). Then, the samples were labelled with a secondary antibody, either horse anti-mouse or goat anti-rabbit biotinylated immunoglobulin (respectively BA-2000 and BA-1000; Vector Laboratories, Inc., Burlingame, CA, USA), and successively conjugated with avidin-biotin peroxidase complex (Vector Laboratories, Inc., Burlingame, CA, USA). The reaction products were visualized using 0.33% hydrogen peroxide and 0.05% 3,3′-diaminobenzidine tetrahydrochloride (DAB), as chromogen (Sigma, St. Louis, MO, USA). The sections were finally counterstained with Carazzi’s Emallumen (blue/violet color, Bio Optica, Milan, Italy), dehydrated, and mounted with DPX for light microscopy detection [45,46]. The immunohistochemical controls were performed by omitting the primary antibody and in isotype matched IgG presence.

### 2.5. Qualitative and Quantitative Analyses

Digitally fixed images of mucosal tunica were analyzed at 200× and 1000× magnification using an optical microscope (Olympus, Hamburg, Germany) equipped with an image analyzer (Image Pro Premier 9.1, MediaCybernetics, Rockville, MD, USA). The immunohistochemical data for TLR4, NF-kB p65, NF-kB p50, COX2, iNOS, TNF-α, IL-1β, Cas-3 and Cas-8 were qualitatively evaluated by a blinder examiner and were expressed as negative (−), very weak (−/+), weak (+), moderate (++) and strong (+++) positivity. Additionally, a blinder examiner evaluated immunostaining also quantitatively, as Integrated Optical Density (IOD) using an image analyzer (Image Pro Premier 9.1, MediaCybernetics, Rockville, MD, USA) [47,48]. The data were pooled to represent a mean value and a statistical analysis was applied to compare the results obtained from the different experimental groups. The analysis was performed on five sections for each sample spacing 70 μm and evaluating six random fields with the same area (52 × 10^3^ μm^2^) per section.

### 2.6. Western Blot Evaluation

Samples of the small intestine (i.e., jejunum) of each of the experimental groups were homogenated in lysis buffer (NaCl 150 mM; EDTA 5 mM, pH 8.0; Tris 50 mM, pH 8.0; Triton X-100 1.0%; sodium deoxycholate 0.5%; SDS 0.1%, dH_2_O) containing a mix of protease inhibitors (Complete Mini, Roche Diagnostics GmbH, Mannheim, Germany). The suspension was centrifugated at 14,000 rpm for 10 min at 4 °C. The resulting supernatant was isolated and the protein content was determined by using the bicinchoninic acid protein assay (Pierce^TM^ BCA Protein assay Kit, Thermo Fisher Scientific, Rockford, IL, USA) to provide precisely equal protein loadings for comparative immunoblots. Samples (100 µg of protein each) were heat-denaturated for 5 min, electrophoresed onto 10% SDS-polyacrylamide gels and electroblottered to nitrocellulose membranes (Amersham^TM^ Protran^®^, GE Healthcare Life Science, Uppsala, Sweden). The membranes were blocked with bovine serum albumin solution for 2 h at room temperature and then probed overnight at 4 °C with the following primary antibodies: mouse monoclonal anti-TLR4 (diluted 1:100; Santa Cruz Biotechnology Inc., Dallas, TX, USA) or mouse monoclonal anti-GAPDH (diluted 1:5000; Sigma-Aldrich, St. Louis, MO, USA). Protein detection was carried out using secondary infrared fluorescent dye conjugated antibodies [47]. The blots were visualized using an Odyssey Fc Imaging System (LI-COR Inc., Bioscience, Lincoln, UK).

### 2.7. Statistical Analysis

Results are expressed as mean ± S.E.M. Statistical significance of differences among the experimental groups for all the markers was evaluated by analysis of variance (one-way ANOVA calculated by Origin ^®^ 7SRI, 1991–2002 OriginLab Corporation, One Roundhouse Plaza, Northampton, MA, USA) corrected by a Bonferroni test with the significance set at *p* < 0.05 for immunohistochemical analysis.

## 3. Results

### 3.1. General Observation

During the experimental study, all forty BTBR and CTR mice remained healthy, readily consuming their daily food, but they gained weight as previously described by our group [3,15] and now reported in Table 1.

### 3.2. Light Microscopy

Hematoxylin and Eosin staining of the GI samples was carried out to visualize the general cytoarchitecture. BTBR mice showed longer intestinal villi in the lamina propria (LP) compared to CTR animals (Figure 1a,b). Notably, we showed in BTBR mice a greater number (although not statistically significant; *p* > 0.05) of cells in the LP; these cells had the morphology of inflammatory cells. This finding confirms the more pronounced inflammatory state present in the intestines of these mice.

### 3.3. Immunohistochemical Immunopositivity of TLR4 and Other Proteins (NF-kB, IL-1β, TNF-α and COX-2)

In the BTBR mice, TLR4 immunoreactivity was moderate (++)/strong (+++) and strong (+++) in the epithelial (EP) and in the LP cells, respectively. TLR4 positivity was expressed in the cytoplasm both in the EP and in the LP cells (Figure 2a). Moreover, the positivity was mainly evident in the brush border membranes of the EP cells. Instead, in the CTR mice, TLR4 immunopositivity showed weak (+) and moderate (++) staining in the cytoplasm of the EP and LP cells, respectively (Figure 2b).

The goblet cells (GC) were negative both in BTBR and in CTR mice (Figure 2a, insert).

If we compare BTBR and CTR mice, TLR4 immunopositivity was more up-regulated in the cytoplasm of both EP and LP cells, even if the difference was not significant (*p* > 0.05). These data have been confirmed by statistical analysis as shown in Figure 2c. These above reported observations were also confirmed by TLR4 western blot analyses (Figure 2d).

After evaluating the findings of TLR4 and its different degree of positivity in BTBR mice compared to CTR animals, we decided to study one of the first proteins linked to this marker as reported by Scalise et al. (2021) [49]. Thus, we evaluated NF-kB (p65, p50 subunits) immunipositivity in the same groups of animals.

We demonstrated that the immunohistochemical immunipositivity of the two NF-kB subunits showed different levels in BTBR and not in CTR mice. In this regard, in BTBR mice, we showed moderate (++) positivity of NF-kB p65 in the cytoplasm of EP cells, in their brush border membranes and in the nuclei of the same cells. The positivity was moderate (++) also in the nuclei of the LP cells (Figure 3a). Results from NF-kB p50 subunit immunostaining showed moderate (++) positivity in the cytoplasm of the EP cells, in their brush border membranes and in their nuclei in BTBR mice; instead, only strong (+++) positivity was observed in the LP cells (Figure 3c). This positivity was very high in LP cells, suggesting that the p50 subunit was easily inducible in these cells compared to the p65 subunit.

For CTR animals, we showed that NF-kB p65 and p50 subunits had weak (+) cytoplasmic staining in the EP cells and in their brush border membranes and in the LP cells; sometimes, some nuclei of EP and LP cells were moderately (++) positive (Figure 3b,d).

The GC cells were negative both in BTBR and in CTR mice as shown for TLR4 positivity.

Comparing BTBR and CTR animals, we observed that in BTBR mice, NF-kB p65 and p50 subunit immunopositivities were higher and statistically significant (*p* < 0.05) in the LP cells and in the EP cells (Figure 3a–d). The quantitative evaluations confirming the semiquantitative analysis and the results are reported in Figure 3e.

After the NF-kB findings, we considered some proteins that are induced following their translocation in the cell nucleus, such as COX-2, IL-1β, TNF-α and iNOS, according to Tinkov et al. (2018) [50].

The immunohistochemical immunopositivity for COX-2 showed moderate (++) and weak (+) positivity in the cytoplasm of the EP cells, in their brush border membranes and in the LP cells for BTBR mice (Figure 4a). GC were not positive (Figure 4a). In CTR animals, the immunohistochemistry for this protein showed very weak (+/−) cytoplasmic positivity in the EP and weak (+) staining in the LP cells. No positivity was observed in the brush border membranes of EP cells and in the GC (Figure 4b).

If we compare BTBR and CTR mice, COX-2 immunostaining was significantly up-regulated in the LP cells (*p* < 0.05) (Figure 4a) in the BTBR mice, but if we consider EP cells, COX-2 positivity was not significatively higher (*p* > 0.05) in either of the two groups (Figure 4a,b). The quantitative evaluations confirming the semiquantitative analysis and the results are reported in Figure 4c.

We then assessed the immunopositivity of IL-1β and TNF-α as other mediators of inflammation in many diseases and in ASD [51,52,53].

In the BTBR mice, immunohistochemistry for IL-1β showed strong (+++) positivity in the cytoplasm of LP cells and weak (+) staining in EP cells. Sometimes, the brush border membranes of EP cells were also weakly (+) positive. In addition, the nuclei of LP cells were strongly (+++) positive in the same groups of animals (Figure 5a). Notably, this result showed that this protein was expressed at a high level and statistically significantly in the LP cells (*p* < 0.05) with respect to the EP cells (Figure 5c). No positivity was observed in the GC (Figure 5a). In CTR animals, IL-1β had a weak (+) immunopositivity in the cytoplasm of the EP and LP cells. Sometimes, the nuclei of the EP and LP cells were weakly (+) or moderately (++) positive (Figure 5a). No positivity was observed in the goblet cells also for CTR.

Comparing BTBR and CTR animals, the statistical analysis showed that the staining of this protein was very high in the LP cells and lower in the EP cells, with a significant difference (*p* > 0.05) but only in LP cells (Figure 5e).

In BTBR mice, TNF-α staining showed strong (+++) and moderate (++) positivity in the cytoplasm of EP and in LP cells and in the brush border membranes of EP cells (Figure 5c). In the CTR mice, the immunopositivity of this protein was very weak (+/−) in the cytoplasm of EP and LP cells and also in the brush border membranes of EP cells (Figure 5d). Sometimes, there were some cells positive in LP, which could be described as a high presence of the endogenous peroxidase (Figure 5d). Thus, if we compare the BTBR and CTR groups, the difference for this staining is statistically significant (*p* < 0.05) (Figure 5e).

In BTBR mice, iNOS immunopositivity followed the same positivity pattern as COX-2. We found moderate/weak (++/+) positivity and strong (+++) positivity for this protein in the cytoplasm of the EP and LP cells, respectively (Figure 6a). In CTR animals, the immunohistochemistry for this protein showed a very weak cytoplasmic (+/−) positivity in the EP cells and a weak cytoplasmic (+) staining in the LP cells, respectively (Figure 6b). No positivity was observed in GC.

If we compare the BTBR mice to CTR group, the iNOS level was significantly up-regulated (*p* < 0.05) in the LP cells of BTBR mice (Figure 6c). On the contrary, if we consider EP cells, the level of iNOS was higher in BTBR mice, but not significatively higher if compared to CTR animals (*p* > 0.05) (Figure 6c).

Additionally, for Cas-3 and Cas-8 immunopositivity, the positivity pattern trend did not change; the immunohistochemical assay highlighted a higher staining in both the EP cells and the LP cells in BTBR mice (data not shown). No positivity was observed in GC (data not shown). If compared to the CTR group, both Cas-3 and Cas-8 stainings were significantly up-regulated (*p* < 0.05) in LP cells of BTBR mice, but if we consider EP cells, the positivity of caspases was not significatively higher if compared with the CTR level (*p* > 0.05). The quantitative evaluations are reported in Figure 7.

## 4. Discussion

The present study is a starting point for other research, since a significant number of ASD children show “dysbiosis” along with immune dysfunction [54,55]. Interestingly, it is known that dysbiosis is often associated with a disruption of the mucosal barrier that is responsible for the alteration in the intestinal permeability leading to a “leaky gut” state [11]. Moreover, the stimulation of dysbiosis could alter the neuronal functions and trigger an autistic behavioral phenotype [55,56,57]. This could become a vicious circle, so it is important to carry out additional research to determine whether dysbiosis has a role also in this disease [54,55].

Our research group has already analyzed in a previous paper the role of the goblet cells in maintaining the intestinal functionality, focusing also on the alterations that could involve these cells in patients affected by the ASD-related dysbiosis [15]. Based on these findings, we performed this study to examine the morphology of the small intestine in BTBR mice but, moreover, to deepen the first results obtained, expanding our analysis, and considering not specific cells, but the more general condition of chronic inflammation to which autistic patients seem to be constantly subjected. For this reason, we decided to evaluate the effects of TLR4 signaling on some proinflammatory and apoptotic markers. The obtained results are: (1) the mucosal tunica showed longer intestinal villi; (2) TLR4 is overexpressed in the mucosal tunica as well as the NF-kB subunits (p65 and p50); (3) the up-regulation of the two previous proteins involve, in turn, many proinflammatory and apoptotic proteins, such as COX-2, IL-1β, iNOS, TNF-α, Cas-3 and Cas-8.

Our morphological results are consistent with the data suggesting that the “leaky gut” state showing longer intestinal villi in BTBR mice is related to the changes in intestinal epithelial morphology. The turnover of these cells may alter digestion, absorption, and downstream processing of nutrients [58]. It is also known that the villi length and related epithelial surface are key factors determining the exchanges of the intestinal mucosa with luminal contents that induce the alterations of microbiota and an increased number of inflammatory cells [59,60,61]. As regards the alterations of microbiota, Puricelli and collaborators (2022) reported that microbial niches, present in the intestinal lumen, interact with the GI mucosa by directly secreting active metabolites or by indirectly stimulating the release of neurotransmitters by enteroendocrine cells [62]. Knowing the important role of these cells and the other components of GI cells [63], we decided to evaluate the possible link between intestinal cells and TLR4, whose immunopositivity is critical to help or regulate the gut microenvironment and the immune system in healthy and pathological states [28]. In BTBR mice, our results showed higher levels in EP and LP cells, but the most important result was finding its immunopositivity also in the brush border membranes of intestinal cells. To our knowledge, this is the first finding showing an increased staining of this receptor in intestinal brush border membranes in an autistic mice model; in this regard, we assume that the up-regulation of TLR4 not only in the EP but also in the brush border may amplify the nutrient absorption, thus favoring weight gain, as we found in BTBR mice.

Of note, TLR4 is an innate immune protein that recognizes glycolipid lipopolysaccharide (LPS), an essential component of the cell wall of all gram-negative bacteria, which are part of the intestinal flora, and always present in the lumen [64]. Accordingly, the alteration of gut microbiota through the modification of LPS could induce an increase of TLR4 as we demonstrated. In terms of the up-regulation of TLR4 in the intestinal LP of BTBR mice, we identified a large number of cells compared to CTR animals and they seem to be inflammatory cells. It is known that ASD shows dysfunction in immune cells and intestinal inflammation [61,65]; therefore, this result runs in parallel with those previously obtained.

We then evaluated other proteins: first at all, we considered NF-kB and, in turn, COX-2, IL-1β, TNF-α and iNOS. Subsequently, we studied two proteins that are important for the apoptotic pathways (Cas-3 and Cas-8). We demonstrated, in EP and LP cells in BTBR mice, an increased positivity of NF-kB in the cytoplasm and in the nuclei in inactive and active form, respectively. This activation regards mainly the p50 subunit in the intestinal LP of BTBR mice; this finding suggests that the p50 homodimer is induced and activated to a greater extent in response to intestinal inflammation, as demonstrated by De Plaen et al. (1998) [66]. In detail, these authors showed that the platelet-activating factor induced NF-kB in the nuclei of intestinal LP cells and that the activated protein contains predominantly p50 subunit. In terms of the localization of NF-kB, our data confirmed its translocation from the cytoplasm to the nucleus, where it activates the expression of specific genes [67]. We demonstrated an increased level of all studied proteins, underlining the above reported data.

The findings also support a link between TLR4 and the NF-kB pathway, starting a cascade that leads finally to the activation of NF-kB and the production of its related proinflammatory and apoptotic proteins [4,61,68,69,70]. In particular, the level of apoptotic proteins, such as Cas-3 and Cas-8, increased due to the oxidative stress induced by these altered pathways [71].

These suggested mechanisms are depicted in the Figure 8.

## 5. Conclusions

In conclusion, our results suggest an important role of chronic inflammation and of TLR4 signaling in the intestinal dysfunction that is typical of ASD patients.

Moreover, even if these are only preliminary outcomes, they allowed us to better delineate the inflammatory background that characterized the clinical presentation. Moreover, we can also better describe which is the pathway activated by TLR4, identifying also the other actors involved, such as NF-kB on one side and caspases on the other.

Much more research is still required into its usefulness as a possible target for therapeutic approaches to reduce intestinal disorders in ASD.

## Figures and Tables

**Figure 1 ijms-23-08731-f001:**
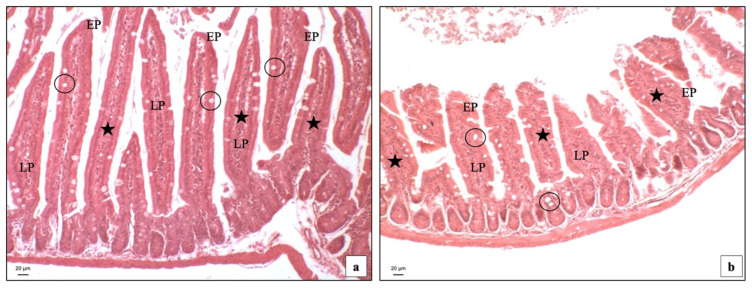
Histological images stained with the hematoxylin and eosin technique, which depict an overview of small intestine samples in BTBR (**a**) and CTR (**b**) mice, 100×. LP: Lamina propria; EP: epithelial cells; black circles: goblet cells; black stars: inflammatory cells. The number of animals was 10 for BTBR and 10 for CTR; moreover, 3 samples for each animal were used. The images are a representative sample of all the examined tissues.

**Figure 2 ijms-23-08731-f002:**
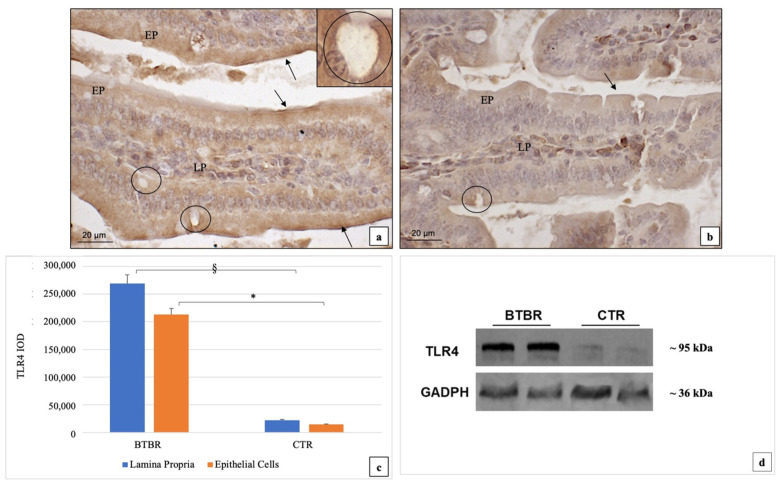
Representative images of TLR4 immunostaining in GI samples of BTBR (**a**) and CTR mice (**b**); 200×. The immunohistochemical signal (marked in brown) could be identified in the cytoplasm both in the EP and in the LP cells. The positivity is mainly evident in the brush border membranes of the EP cells (highlighted by a black arrow). The goblet cells were negative both in BTBR and in CTR mice; they are illustrated in the insert of (**a**) (1000×). Statistical analyses of TLR-4′s levels both for LP and EP in BTBR and CTR animals are reported in graphic in (**c**). Data obtained by Western blot showing TLR4 levels in BTBR and CTR small intestines are reported in (**d**). Data are presented as mean ± SD. * *p* < 0.05 TLR4 immunopositivity in EP, BTBR vs. CTR mice; § *p* < 0.05 TLR4 immunopositivity in LP, BTBR vs. CTR mice. LP: Lamina propria; EP: epithelial cells; black circles: goblet cells; black arrows: brush border membranes. The number of animals was 10 for BTBR and 10 for CTR; moreover, 3 samples for each animal were used. The images are a representative sample of all the examined tissues.

**Figure 3 ijms-23-08731-f003:**
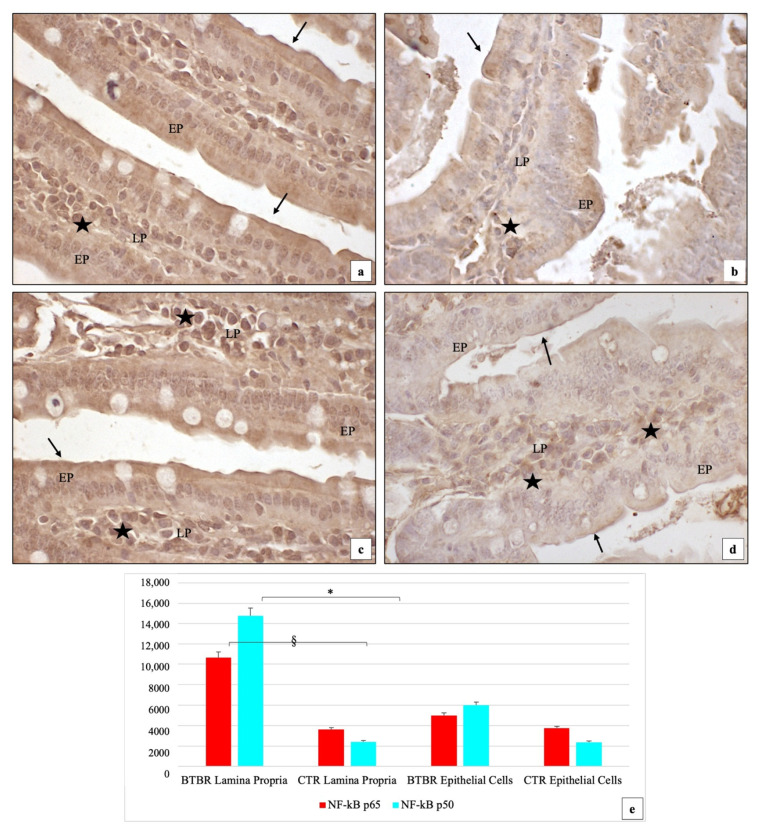
Representative image of NF-kB p65 and p50 immunostaining in gut samples of BTBR and CTR mice ((**a**) and (**b**), respectively, for subunit p65 and (**c**,**d**) for subunit p50); 200×. Statistical analyses of NF-kB p65 and p50 immunopositivity both for LP and EP in BTBR and CTR animals are reported in graph (**e**). Data are presented as mean ± SD. § *p* < 0.05 NF-kB p65 positive cells in LP, BTBR vs. CTR mice; * *p* < 0.05 NF-kB p50 positive cells in LP, BTBR vs. CTR mice. LP: Lamina propria; EP: epithelial cells; black circles: goblet cells; black stars: inflammatory cells; black arrows: brush border membranes. The number of animals was 10 for BTBR and 10 for CTR and 3 samples were used for each animal. The images are a representative sample of all the examined tissues.

**Figure 4 ijms-23-08731-f004:**
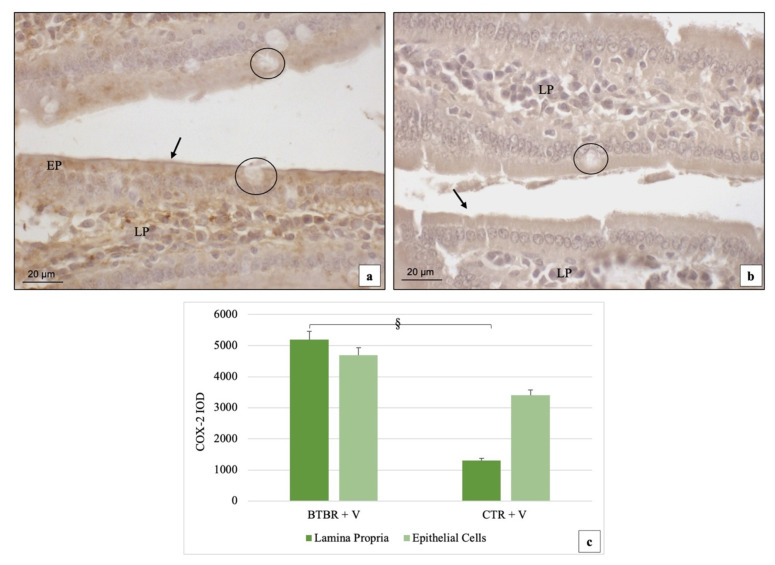
Representative image of COX-2 immunostaining in gut samples from BTBR and CTR mice ((**a**) and (**b**), respectively); 200×. Statistical analyses of COX-2 staining for both LP and EP in BTBR and CTR animals are reported in graph (**c**). Data are presented as mean ± SD. § *p* < 0.05 COX-2 positive cells in LP, BTBR vs. CTR mice. LP: Lamina propria; EP: epithelial cells; black circles: goblet cells; black arrows: brush border membranes. The number of animals was 10 for BTBR and 10 for CTR and 3 samples were used for each animal. The images are a representative sample of all the examined tissues.

**Figure 5 ijms-23-08731-f005:**
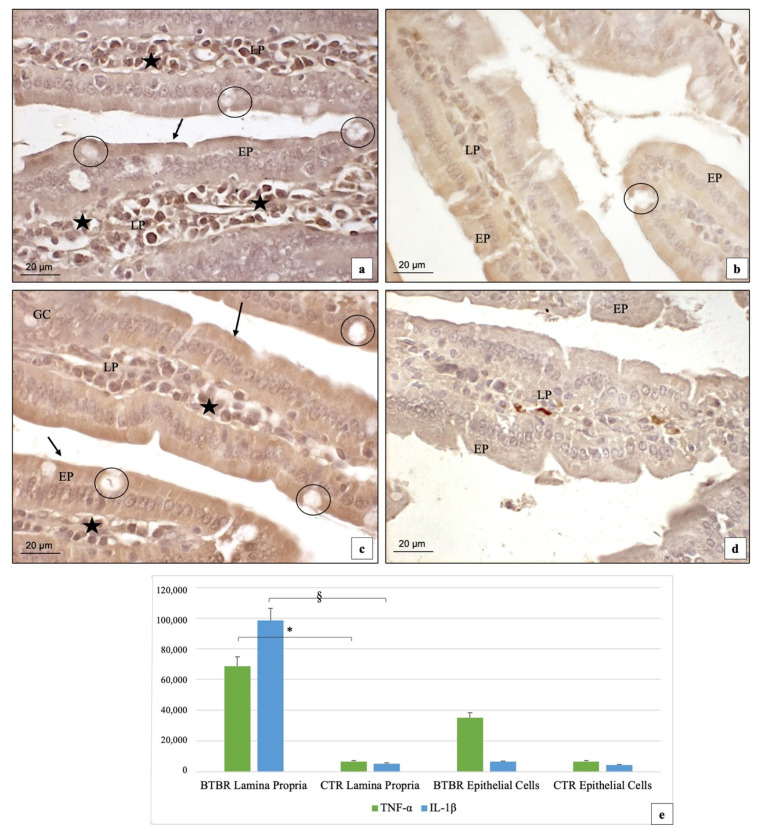
Representative image of IL-1β and TNF-α immunostaining in gut samples of BTBR and CTR mice ((**a**) and (**b**), respectively, for IL-1β and (**c**,**d**) for TNF-α). The immunohistochemical signal (marked in brown) of IL-1β was strong in the cytoplasm of LP cells (LP) and weak in EP cells (EP) in BTBR mice; 200×. Quantitative analyses are presented as mean ± SD in graph (**e**). § *p* < 0.05 IL-1β positive cells in LP, BTBR vs. CTR mice; * *p* < 0.05 TNF-α positive cells in LP, BTBR vs. CTR mice. LP: Lamina propria; EP: epithelial cells; black circles: goblet cells; black arrows: brush border membranes; black stars: inflammatory cells. The number of animals was 10 for BTBR and 10 for CTR and 3 samples were used for each animal. The images are a representative sample of all the examined tissues.

**Figure 6 ijms-23-08731-f006:**
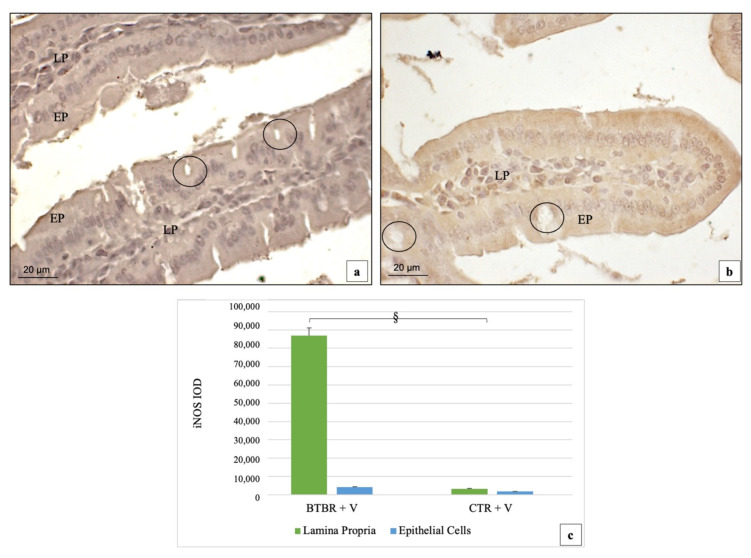
Representative image of iNOS immunostaining in gut samples of BTBR (**a**) and CTR mice (**b**); 200×. Quantitative analyses are presented as mean ± SD in graph (**c**). § *p* < 0.05 iNOS positive cells in LP, BTBR vs. CTR mice. LP: Lamina propria; EP: epithelial cells; black circles: goblet cells. The number of animals was 10 for BTBR and 10 for CTR; moreover, 3 samples for each animal were used. The images are a representative sample of all the examined tissues.

**Figure 7 ijms-23-08731-f007:**
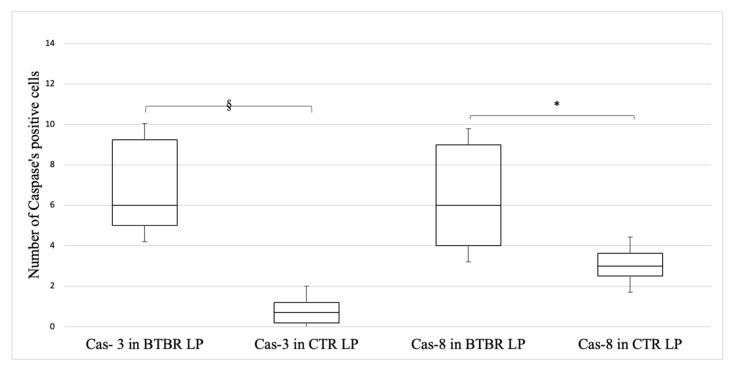
Quantification of Cas-3 and Cas-8 positive cells in the lamina propria (LP) of both BTBR and CTR mice. Data are presented as mean ± SD. § *p* < 0.05 number of Cas-3 positive cells, BTBR vs. CTR mice; * *p* < 0.05 number of Cas-8 positive cells, BTBR vs. CTR mice. The number of animals was 10 for BTBR and 10 for CTR and 3 samples for each animal were used. The images are a representative sample of all examined tissues.

**Figure 8 ijms-23-08731-f008:**
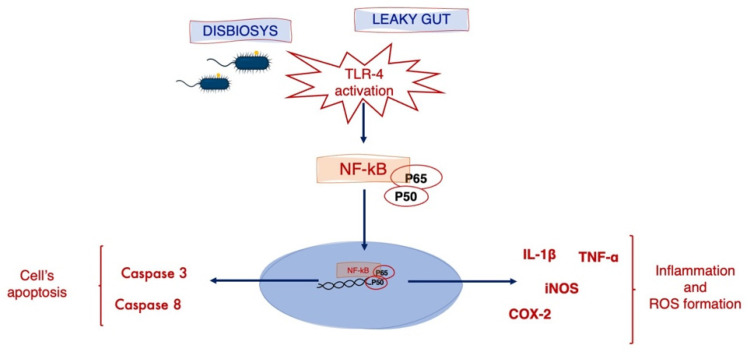
Schematic representation of the link between TLR4 and the NF-kB pathway that turns on an intracellular cascade that leads finally to the activation of NF-kB and the production of its related proinflammatory and apoptotic proteins, such as COX-2, IL-1β, TNF-α and iNOS from one side and Cas-3 and Cas-8 from the other.

**Table 1 ijms-23-08731-t001:** Body weights (g) of BTBR and CTR mice at the beginning and at the end of the experimental period.

Age of BTBR and CTR Mice	BTBR Mice (Mean ± SD, g)	CTR Mice (Mean ± SD, g)
3 weeks of life	19.6 ± 1.41	10.45 ± 0.95
13 weeks of life	34.29 ± 2.3	27.19 ± 1.46

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
