# Peer review of "Impairment in the Intestinal Morphology and in the Immunopositivity of Toll-like Receptor-4 and Other Proteins in an Autistic Mouse Model"

_ijms, 2022, doi:10.3390/ijms23158731_

Round 1

Reviewer 1 Report

In general, the concept of this work is really interesting. But there are some points of criticism which need to be clarified before the publication

Title  – Title is misleading. From anatomical point of view gastrointestinal tract (also known as alimentary tract or digestive tract) is a tract which passes food from mouth to the anus. I have the impression that the authors limit the GI tract to segments located distally from the stomach.

Line 20 and throughout the text – the term expression should be restricted to genes only

Line 72 – TNF-α stands for “tumor necrosis factor” not nuclear !

Line 96 – please write ad libitum in italic.

Line 109 – it is of importance what segment of the small intestine was dissected out (duodenum, ileum or jejunum).

Line 131 – 140 – please provide manufacturer’s codes for antibodies used.

Line 148 - The major problem of this study is that the authors did not test properly the specificities of antibodies they used. The preadsoprtion tests are necessary.

Line 151 – Immunohistochemistry is qualitative but not quantitative method. Could authors address this ?

Figure 1-6 – scale bars are missing

Line 213 – the presence of TLR4 was previously found on the surface of the mammalian small intestine enteric neurons (Arciszewski MB, Sand E, Ekblad E. Vasoactive intestinal peptide rescues cultured rat myenteric neurons from lipopolysaccharide induced cell death. Regul Pept. 2008 Feb 7;146(1-3):218-23. doi: 10.1016/j.regpep.2007.09.021.). First, this work should be cited and discussed. Second, it is of interest for the readers did the authors observe the presence/changes of TLR4 in the enteric nervous system of the studied animals.

Line 448 – conclusion in the present form is very weak and trivial. Please provide in a separate chapter clear-cut conclusions.

Line 493 – please verify the correctness of this reference

Author Response

  • We thank the Referee for the suggestion; we changed the title according to the indications (lines 2, 20).
  • We thank the Referee for the indication: w e changed the term throughout the text (lines 20, 24, 25, 84, 155, 252, 255, 272, 275, 286, 287, 288, 289, 290, 298, 301, 304, 312, 322, 328, 336, 343, 351, 355, 361, 377, 378, 381, 383, 385, 395, 396, 398, 400, 441, 443, 444, 445, 453, 462, 470, 474).
  • We thank the Referee for the annotation: we corrected the mistake in the text (line 76).
  • We thank the Referee for the annotation: we changed the font according to the indication (line 100).  
  • We thank the Referee for the annotation: we better specified in the Material and Methods section (lines 113,199) which part of the intestinal tract we have analyzed. We have also added a schematic representation in order to better identify the region that we’ve considered (Fig.1).
  • We thank the Referee for the suggestion: we better specified in the Material and Methods the antibodies codes (lines 163-174).
  • We thank the Referee for the indication: we better specified in the Material and Methods also the secondary antibodies features and codes (line 173).
  • We thank the Referee for the suggestion: we have better indicated in the Material and Methods that we have analyzed the immunohistochemical data for TLR4, NF-kB p65, NF-kB p50, COX2, iNOS, TNF-α, IL-1β, Cas-3 and Cas-8 not only qualitatively [as negative (-), very weak (-/+), weak (+), moderate (++), and strong (+++) positivity], but also as Integrated Optical Density (IOD) obtained by an image analyser (Image Pro Premier 9.1, MediaCybernetics, Rockville, MD, USA) (lines 184-192). Moreover, we also changed the title of the paragraph in the Material and Methods section (line 183).
  • We thank the Referee for the suggestion: we added the scale bars in each figure (Fig.2-7).
  • We thank the Referee for the suggestion: we added the specific bibliography regarding the physiological role of the enteric nervous system (lines 43-45), however, although some data already state that there is also an involvement of the SNE, we preferred to focus on the condition of epithelial cells as they are also the body's first defense and little is known about the effect that oxidative stress and chronic inflammation may have on dysbiosis and intestinal function. It would, however, be our intention to study this further in a second moment. We added some considerations about this point at the end of the Discussion (lines 422-424).
  • We thank the Referee for the suggestion: we added a final summary paragraph summarising what has been achieved (lines 492-500).
  • We thank a lot the Referee for the suggestion: we checked and corrected the reference (line 546).

Reviewer 2 Report

In the manuscript entitled: “Impairment in the gastrointestinal morphology and in the expression of Toll-like receptor-4 and other proteins in an autistic 3 mouse model” authors described increased expression of TLR4, NF-kB, and several inflammatory molecules in the small intestine of mouse model of autism spectrum disorder (ASD) as compared to control mice. Authors confirmed previous findings of other researchers in the field and suggest the use of TLR4 and other inflammatory molecules to be targeted to develop therapeutic interventions for gastrointestinal problems/symptoms often noted in ASD patients.

There are some concerns about the data evaluation and other minor concerns listed below:

1) Mice were maintained at 20°C which is somewhat cooler temperature for mice.

2) What is the composition of lysis buffer?

3) Table 1: How are weights of animals expressed? What is the unit value of weight?

4) How many animals were used in the study?

5) Please include magnification bar for all images.

6) In Figure 2, lamina propria shows very diffuse TLR4 staining as compared to discreet cells in the control tissue yet histogram shows higher levels of TLR4 in BTBR tissue. In the same figure apical part of epithelial cells in BTBR mice has intense staining as compared to control tissue. The staining pattern (above) seems to match with Western blot data but not with the histogram (Figure 2c). Can you please explain the discrepancy?

7) It will useful to readers if authors can provide a diagram showing the part of the small intestine used for histology and GI part used for Western blotting.

8) Figure 5e shows higher staining in lamina propria for TNFalpha as compared to IL-1beta. This data does not appear to match with images in Figure 5a and 5c. Cells in the lamina propria appears quite distinct. Please consider representing data as intensity of immunostaining in cells rather than as LP.

9) Mucosa and circular folds/villi both have increased size in ASD model mice. What is its significance?

10) The goblet cells are not affected suggesting they are secreting mucin to protect the epithelium. Did authors correlate the number of globlet cells with inflammation observed in BTBR mice small intestine? The goblet cells are also involved in immunoregulation. It may be thus important to learn if their number is altered in ASD.

Author Response

  • We thank the Referee for the annotation: the mice were kept at that specific temperature according to the instructions given to us by the veterinary in charge of the university's animal enclosure, in accordance with the recommendations of the Animal Welfare Commission of 18 June 2007 (Guidelines for the accommodation and care of animals used for experimental or other scientific purposes).
  • We thank the Referee for the suggestion: we better specified in the Material and Methods section (lines 200-202) the detail that has been required.  
  • We thank the Referee for the suggestion: we better specified in the table’s description (line 229) the detail that has been required.  
  • We thank the Referee for the suggestion: we better specified in the Results section (line 226) the detail that has been required.  
  • We thank a lot the Referee for the suggestion: we added the scale bars in each figure (Fig. 2-7).
  • We thank a lot the Referee for the annotation: we have reanalyzed the data obtained from the immunohistochemistry and remodulate the histograms according to them (Fig.3).
  • We thank the Referee for the annotation: we better specify in the Material and Methods section (lines 113,199) which part of the intestinal tract we have analyzed; we have also added a schematic representation in order to better identify the region that we’ve considered (Fig.1).
  • We thank the Referee for the annotation: we have reanalyzed the data obtained from the immunohistochemistry and remodulate the histogram according to them (Fig.6). Now, these results are similar to that observed by qualitative analysis.
  • We thank the Referee for the comment: from a morphological point of view, we appreciated the difference. This could explain the alterations in metabolism and absorption that characterize autistic patient. It would in any case deserve more in-depth study to be able to draw more specific conclusions, also associating the composition of the intestinal microbiota.
  • We thank the Referee for the suggestion: we better specified in the text the reasons of our choice (lines 418-420).

Round 2

Reviewer 1 Report

Most of my comments were taken into acoount.

Some still need to be clarfied.

1. The correct abbreviation for the enteric nervous system is ENS (not SNE as used in line 44)

2. Figure 1 is misleading. In line 113 the authors describe tissue sampling (from mice) and refer in figure 1 to human GIT !!! These two have nothing to do each other. Figure 1 is redundant or should be replaced with the schema of the mouse GIT.

3. "Line 148 - The major problem of this study is that the authors did not test properly the specificities of antibodies they used. The preadsoprtion tests are necessary." - not answered!

Author Response

We thank the referee and according to suggestions we  corrected the English term (line 44: ENS) and we deleted Fig.1.

We apologize to the referee for the non-response. We have better emphasized the preabsorption test by pointing out the literature we used for the test (lines 134-136).

Reviewer 2 Report

In the manuscript entitled: “Impairment in the intestinal morphology and in the immunopositivity of Toll-like receptor-4 and other proteins in an autistic 3 mouse model” authors described increased expression of immunoreactive TLR4, NF-kB, and several inflammatory molecules in the small intestine of mouse model of autism spectrum disorder (ASD). Authors demonstrated an increase in inflammatory proteins in intestine of ASD mice using immunohistochemical and Western blot techniques. In addition, authors identified pathway(s) involved in the activation of TLR4. Their data confirms previous findings of other researchers in the field as well as their own studies in the field. Identification of microbiota and alteration of intestinal morphology/histology, and activation of TLR4 and other inflammatory molecules may be crucial to develop targeted therapies for ASD patients.

Authors have addressed all the concerns.

Author Response

We thank the referee's for the evaluation.